# Fabrication of Dimensional and Structural Controlled Open Pore, Mesoporous Silica Topographies on a Substrate

**DOI:** 10.3390/nano12132223

**Published:** 2022-06-28

**Authors:** Tandra Ghoshal, Atul Thorat, Nadezda Prochukhan, Michael A. Morris

**Affiliations:** School of Chemistry, AMBER and CRANN, Trinity College Dublin, D02 AK60 Dublin, Ireland; atulthorat007@gmail.com (A.T.); prochukn@tcd.ie (N.P.)

**Keywords:** open mesopores, silica channels, block copolymers, hard mask, etching

## Abstract

Open pore mesoporous silica (MPS) thin films and channels were prepared on a substrate surface. The pore dimension, thickness and ordering of the MPS thin films were controlled by using different concentrations of the precursor and molecular weight of the pluronics. Spectroscopic and microscopic techniques were utilized to determine the alignment and ordering of the pores. Further, MPS channels on a substrate surface were fabricated using commercial available lithographic etch masks followed by an inductively coupled plasma (ICP) etch. Attempts were made to shrink the channel dimension by using a block copolymer (BCP) hard mask methodology. In this regard, polystyrene-*b*-poly(ethylene oxide) (PS-*b*-PEO) block copolymer (BCP) thin film forming perpendicularly oriented PEO cylinders in a PS matrix after microphase separation through solvent annealing was used as a structural template. An insitu hard mask methodology was applied which selectively incorporate the metal ions into the PEO microdomains followed by UV/Ozone treatment to generate the iron oxide hard mask nanopatterns. The aspect ratio of the MPS nanochannels can be varied by altering etching time without altering their shape. The MPS nanochannels exhibited good coverage across the entire substrate and allowed direct access to the pore structures.

## 1. Introduction

Ordered mesoporous materials have attracted attention since their discovery by Mobil Oil Research and Development scientists in 1992 [1,2]. Mesoporous silica (MPS) thin films have promising applications in microelectronics, sensing, catalysis, separation and optoelectronics [3,4,5,6,7,8,9]. The evaporation-induced self-assembly (EISA) method, developed by Lu et al. in 1997 [10], has become the most convenient method to generate these films. Briefly, at casting the film at a concentration of the solution below the critical micelle concentration (CMC) no micelles are formed [11]. As the solvent evaporates, the film concentration exceeds the CMC, micelles will form and as higher concentrations are developed, the micelles can stack forming an ordered structure. The ordered mesoporous silica film is formed around the micellar template and can be obtained through drying/calcination to remove the organic surfactant [12,13,14]. The structure, morphology, periodicity is dependent on variables such as reactant concentration, synthesis temperature, pH of solution and the nature of surfactant [12].

However, a limitation of these materials is that they are ‘closed’ (the pores running in the surface plane) and inaccessible from the surface. For practical exploitation of these high-tech materials (in application such as microelectronics, catalysis, photonics, chip-based sensing, etc.) pore access is required and the development of micro- and nanofabrication patterning technologies to expose the pores are required [15]. Here, patterning is defined as creating topographical arrangements that reveal the pores. Various patterning techniques have been proposed for mesostructured films. They are often based on established methods, including conventional and less conventional lithographic techniques [15], UV patterning [16], rapid prototyping by micropen lithography [17,18], dip-pen nanolithography [19,20], site-selective deposition on surfaces pre-patterned with self-assembled monolayers [21], electron-beam lithography [22,23] and inkjet printing [24]. These methods are mostly restricted to flat surfaces only and can be challenging compared to etching of dense films.

Block copolymers (BCP) self-assembly has potential use in the nanofabrication of memory and semiconductor devices due to the ease of processing, inexpensive cost and simple integration capability [25,26]. BCP lithography offers an attractive alternative patterning technology to conventional lithography since the BCPs can self-assemble on length scales from a few to tens of nanometers [27]. Spin-cast followed by solvo-thermal treatment is a simple approach that can be applied to generate vertically oriented cylindrical microdomains through the formation of solvent fronts and/or alteration of interfacial chemistry. It is also necessary to adapt a simple, cost effective method to convert them into the material patterns in terms of identical size/shape, regularity where each individuals with same composition.

This paper describes the synthesis of thin films of MPS with a two-dimensional (2D) hexagonal structure using TEOS (tetraethylorthosilicate) and Pluronic^TM^ triblock copolymer surfactants. Variation in mesopore sizes and the film thicknesses of MPS thin films with respect to concentrations of precursors has been studied. MPS channels was achieved by patterning the film with a commercial available lithographic mask followed by ICP etching. Smaller dimensional MPS nanochannels was fabricated by using ‘insitu’ BCP hard mask methodology.

## 2. Experimental

### 2.1. Materials

TEOS (≥99.999%), 0.2 M HCl, anhydrous ethanol (≥99.9%), oxalic acid dihydrate (C_2_H_2_O_4_, 2H_2_O) and iron (III) nitrate nonahydrate (Fe(NO_3_)_3_, 9H_2_O) were purchased from Sigma-Aldrich and used as received. All Pluronic surfactants were purchased from BASF and used as received. A polystyrene-*b*-poly(ethylene oxide) (PS-b-PEO) diblock copolymer was purchased from Polymer Source and used without further purification (number-average molecular weight, M_n_, PS = 42 kg mol^−1^, M_n_, PEO = 11.5 kg mol^−1^, M_w_/M_n_ = 1.07, M_w_: weight-average molecular weight).

### 2.2. Synthesis of Mesoporous Silica Thin Films

Mesoporous silica thin films were prepared using triblock copolymers (Pluronics) on silicon substrates by following a variation in the procedure reported previously [28]. The procedure to synthesis the mesoporous silica thin film is shown in Figure 1. 2.08 g of TEOS, 3 g of 0.2 M HCl, 1.8 g water and 5 mL anhydrous ethanol were mixed and heated at 60 °C for 1 h in temperature-controlled preheated oven. This solution was allowed to cool to room temperature. 15 mL of a 5 wt% of Pluronic surfactant and 10 mL anhydrous ethanol were added with vigorous stirring. Silicon substrates were coated using this solution at 3000 rpm for 30 s. These silicon substrates were then calcined at 450 °C for 2 h at a ramp rate of 1 °C min^−1^.

### 2.3. Synthesis of Mesoporous Silica Channels Using Lithographic Masks

The procedure to generate mesoporous silica channels using lithographic masks is shown in Figure 2. Commercial lithographic resist materials such as SU-8 2000 was used to generate the mesoporous silica channels. SU-8 2000 is a commonly used epoxy-based negative photoresist originally developed at IBM [29]. Firstly, SU-8 2000 was spin coated over a mesoporous silica thin film (30 s at 2000 rpm). The thickness of photoresist can range from below 1–300 µm and still be processed with standard contact lithography. It is used to pattern high aspect ratio (>20) structures. To obtain vertical sidewalls in the SU-8 2000 resist, a long pass filter was used to eliminate UV radiation below 350 nm. Strong agitation was applied while developing thick film structures to improve the etching quality which reduces anisotropic etching and surface roughness and simultaneously increases the etching rate. Then this photoresist is ICP dry etched for specific time to fabricate the mesoporous silica topographies.

### 2.4. Synthesis of Mesoporous Silica Channels Using In Situ Hard Mask BCP Approach

Figure 3 and Figure 4 illustrate the process flow diagram of the fabrication of ordered MPS nano-channels by a BCP assisted approach. The synthesised MPS thin film on a silicon substrate is shown in Figure 4A. The PS-*b*-PEO (42K-11.5K) thin film was fabricated by spin coating the polymer solution at 3000 rpm for 30 s on the as-synthesized mesoporous silica film. The film was exposed to a toluene/water (50:50, *v*/*v*) mixed vapour placed at the bottom of a closed vessel kept at 50 °C for 1 h to induce chain mobility and allow microphase separation to occur (Figure 3a and Figure 4B). Separate reservoirs were used for each solvent to avoid azeotropic effects. The resultant phase separated film was immersed in ethanol at 40 °C for 15 h to partially modify the PEO component causing ‘activation’ of the cylinders (Figure 3b and Figure 4C). The film was dried under nitrogen. For the fabrication of oxide nanodots, iron (III) nitrate nonahydrate (Fe(NO_3_)_3_, 9H_2_O) precursor was used. 0.4 wt% of iron nitrate was dissolved in ethanol and spin coated onto the nanoporous film (Figure 3c and Figure 4D). UV/ozone treatment was carried out to convert the precursor into iron oxide as well as for complete degradation of the residual polymers (Figure 3d and Figure 4E). Iron oxide nanodots remain on the top of mesoporous silica film.

### 2.5. Plasma Etch Pattern Transfer

These iron oxide nanodot arrays were used as a hard mask for pattern transfer onto the substrate. Pattern transfer was accomplished using an STS, Advanced Oxide Etch (AOE) ICP etcher. The system has two different RF generators, one to generate and control the plasma density by direct connection to the antenna coil, while the other one was used to adjust and control the energy of ions by connecting it to the substrate holder. During etching, the sample was thermally bonded to a cooled chuck (10 °C) with a pressure 9.5 Torr. For the oxide etch, the process parameters were optimised to a C_4_F_8_/H_2_ gas mixture (21 sccm/30 sccm) using an ICP coil power of 800 W and a Reactive Ion Etching (RIE) power of 80 W. MPS nano-channels having nanodots on the top were formed by ICP dry etching for 20 s (Figure 3e and Figure 4F) using the iron oxide as a hard mask. The height of the mesoporous silica features was varied by simply varying the silica etch time. For the removal of iron oxide nanodots, the substrate was immersed into 10 wt% aqueous solution of oxalic acid dihydrate (C_2_H_2_O_4_ 2H_2_O) for 2 h at room temperature followed by washing with water and drying of substrates (Figure 3f and Figure 4G).

### 2.6. Characterization

X-ray diffraction (XRD) patterns were recorded on a PANalytical MPD instrument using an Xcelerator detector and a Cu Kα radiation source at a working power of 45 kV and 40 mA. BCP film thicknesses were measured using a spectroscopic ellipsometer “Plasmos SD2000 Ellipsometer” at a fixed angle of 70 ° at a minimum of five different locations on the sample. Average values were reported as the measured thickness value. A two-layer β-spline model (SiO_2_ + BCP) was used to simulate experimental data. Top-down and cross-sectional SEM images of samples were obtained by a high resolution Field Emission Zeiss Ultra Plus-scanning Electron Microscope (SEM) operating at 10 kV. Samples were prepared for transmission electron microscopy (TEM) cross sectional imaging with an FEI Helios Nanolab 600i system containing a high resolution Elstar™ Schottky field-emission SEM and a Sidewinder FIB column and TEM was carried out on a JEOL JEM 2100 microscope operated at a voltage of 200 kV.

## 3. Results and Discussion

### 3.1. Silica Mesoporous Thin Films Formation Using Different Pluronics

To form uniform and well-ordered silica mesopores, the concentration of silica precursor i.e., TEOS was carefully optimized [30,31]. Figure 1 shows the low angle XRD patterns of MPS thin films synthesized using 0.01, 0.005 and 0.0033 M concentration of TEOS respectively in the presence of Pluronic P-123 surfactant. The as-synthesized MPS thin films using 0.01 M TEOS predominantly exhibits the (100) reflection peak, the 2nd (200) and 3rd (300) order reflections at 1.54°, 2.95° and 4.42° respectively (Figure 1a) and these indicates a high degree of long range ordering in the MPS thin films. The absence of 3rd (300) order reflections, broadening of the main peaks and presence of additional multiple peaks were seen in mesoporous silica thin films prepared using 0.005 and 0.0033 M TEOS as shown in Figure 1b,c respectively, suggesting less structural ordering in the mesoporous silica thin films with lower TEOS concentrations [32,33,34,35]. The absence of a (110) reflection indicates that the porous arrangements within the films is 2D hexagonal and pores are parallel to the surface plane for 0.01 M TEOS with the P-123 system.

The pore sizes and spacing in mesoporous silica thin films also depends on the type of Pluronic used for the synthesis. Different Pluronics P-123, P-85 and P-65 were used to form the MPS thin films. Each surfactant corresponds to differing molecular weights, used to vary the pore sizes due to the specific micellar arrangement in solution [34,35]. Figure 2 shows the XRD patterns of synthesized MPS thin films using P-123, P-85 and P-65. MPS thin films synthesized using Pluronic P-123 and P-85 with 0.01 M TEOS exhibits a main (100) peak, 2nd (200) and 3rd (300) order reflections at 1.54°, 2.95°, 4.42° and 1.86°, 3.62°, 5.51° respectively, and again indicate the high degree of long range ordering present in the MPS thin films. However, the shift in peak positions for mesoporous silica thin films synthesized using P-85 towards the high angle direction confirms a decrease in pore spacing. Whereas, broadening of the main peak and the absence of 2nd and 3rd order reflections indicates less structural ordering in the MPS thin films synthesized using P-65. The absence of the (110) reflection indicates that the porous arrangements within the films is 2D hexagonal and pores are parallel to the surface plane for P-123 and P-85 systems. The pore sizes of all of the synthesized MPS thin films were calculated from the low angle XRD patterns. The diameter of the pores for P123 with different TEOS concentrations and also for P85 and P65 were calculated from Scherrer formula.
$$D=\frac{0.9\;\lambda }{\beta \;\cos\theta }$$
where D is the diameter of the pores in nm, *λ* is the X-ray wavelength in nm, *β* is the full width half maximum (FWHM) of the diffraction peak in radians and *θ* is the angle of diffraction in degrees. The film thickness, pore sizes and peak positions of mesoporous silica thin films with varying experimental parameters are summarized in Table 1.

### 3.2. Morphological Study of Mesoporous Silica Thin Films by SEM

The morphology and pore size of MPS thin films were determined from the SEM studies. Figure 3a shows that cross-sectional SEM of mesoporous silica thin film obtained using Pluronic P-123. The SEM images confirm XRD observations that mesoporous films have a 2D hexagonal structure with the pores lying parallel to the substrate plane. The mesopores have diameter within the range of 13–15 nm. The thickness of film was found to be 100 nm. Furthermore, these films showed no evidence of structural deformation either at the film surface or at the film-substrate interface indicating good adhesion with the substrate.

Figure 3b shows the cross-sectional SEM images of MPS thin films formed on a topographically defined trench patterned silica substrate. The alignment of nanopores within the channel was affected by using low aspect ratio (channel width to depth) trenches of Si/SiN substrates previously reported by us [36]. Wu et al. have also exploited this technique for aligning MPS films for use as resist moulds [37]. These alignment methods can be improved by a physically modified substrate guiding the long-range ordering of the MPS system. Such an approach has been successfully used to control macroscopic ordering of colloidal spheres and block copolymer films [38,39]. In our case, the MPS thin films prepared on a trenched substrate showed less ordered arrangement of the pores than on planar substrates (Figure 3b). Poor ordering and defects (shape, size and spacings between the pores) can be related to several reasons. These includes non-uniform width of the trench throughout the depth, incommensurability of the pore spacing and complex interfacial interactions of the trench wall with the precursor material during pore development.

Figure 3c shows the cross-sectional SEM of the mesoporous silica thin films with hexagonally arranged pores synthesized using Pluronic P-85. The average diameter of mesopores was found to be 7 nm. The thickness of film was found to be around 95 nm. These films also showed no evidence of structural deformation either at the film surface or at the film-substrate interface as well as exhibiting good adhesion to the substrate. It is also important to note that in all studies, the upper surface of the film is dense silica and pores are generally not present. This is typical for these films.

### 3.3. Fabrication of MPS Channels Using Lithographic Masks

Figure 4 shows the cross-sectional SEM of MPS channels synthesized using a lithographic mask of resist SU-8 2000. SU-8 can be processed with a number of patterning techniques to render high-aspect-ratio and 3D submicron structures. The irradiation source and configuration used for processing determines the maximum lateral resolution, aspect ratio and geometrical complexity of the patterned features [40]. To fabricate mesoporous silica channels, the mesoporous silica thin film surface is partially protected using lithographic resist SU-8 2000 as an etch mask, followed by ICP dry etching. The depth and width of the channels can be varied using different dimensional lithographic resist as well as by varying the etching time. Figure 4a shows ICP dry etched mesoporous silica channels having the lithographic resist on the top. Figure 4b shows the magnifying image of etched and unetched (protected below the lithographic resist) MPS thin film. The thickness of the unetched part is 100 nm. After 10 s ICP dry etching, film thickness decreased to approximately 50 nm. After the etching process, the lithographic resist can be easily removed using mild oxalic acid solution. Figure 4c shows the 10 s ICP dry etched and unetched mesoporous silica thin film after removing lithographic resist. A magnified image of the ordered mesopores are shown in inset of Figure 4c. All of the images suggests that MPS channels can be fabricated by ICP etch process using a resist mask without hampering the hexagonally arranged mesopore structure.

### 3.4. Fabrication of Mesoporous Silica Channels Using In Situ Hard Mask BCP Approach

The standard lithographic mask appeared to indicate good ability to retain the ordering of the mesoporous film and prevent the pore structure from collapsing during etching. But with conventional lithographic methods, it is challenging to shrink the channel dimensions to the size of a few mesopore diameters. The continual reduction of critical dimensions of advanced electronic devices challenges conventional ultraviolet (UV) lithography and requires the use of new and alternative patterning techniques such as double or triple patterning to create substrate features for use in both logic and memory device and interconnect level circuitry which has proven to be slow, complex and expensive [41]. While the lithographic mask can create features down to few tens of nanometers, BCP lithographic technique is capable to create feature as small as sub-5 nm depending on the type and molecular weights of BCPs. For proper validation of the film robustness it is necessary to prepare small features since these are consistent with modern manufacturing dimensions. In this study, an approach to create channels with smaller dimension is attempted using block copolymer lithography. The successful integration of BCP methods into the device requires ultimate control of the self-assembly and the pattern transfer onto the underlying material. However, etching becomes complicated when feature sizes reduce and etch limitations of BCP patterns can lead to silicon features of low aspect ratio and high line edge roughness (LER). To overcome this barrier, we have generated a ‘hard mask’ material using our established insitu inclusion method [42,43].

Figure 5a shows an AFM image of the PS-b-PEO thin film demonstrating the vertically orientated hexagonal arrangement of PEO cylinders inside the PS matrix. The long-range ordering and perpendicular cylinder orientation were formed by annealing [44,45]. The spin coated film in mixed toluene–water environment at a temperature of 50 °C for 1h which induces microphase separation. In the AFM image, darker contrast corresponds to PEO cylinders. The measured average centre-to-centre cylinder spacing is ~42 nm and PEO cylinder diameter is ~19 nm. The SEM image in Figure 5b also represents long range ordering of the PS-b-PEO thin film.

These films were used as a template to prepare iron oxide hard etch mask as reported previously by our group [42,43,44,45]. PEO microdomains are the preferred site to incorporate metal ions into the template. This is realised by an etching and/or modification of the PEO site through immersing the film in anhydrous ethanol at 40 °C for 15 h. Structural arrangement, dimensions and ordering remain unchanged after the treatment [46]. The AFM image (Figure 5c) shows some increase in the phase contrast after this treatment indicating etched/modified PEO microdomains. Also, the SEM image contrast was enhanced by ethanol exposure as seen in Figure 5d reveals porous structure. No thickness change of the polymer film was observed after the ethanol treatment.

Iron oxide nanofeatures were formed by insitu inclusion of iron ions by spin coating the metal nitrate ethanolic solution into nanoporous BCP template. The PEO cylinders (diameter of ~20 nm and depth ~28 nm) can be considered are selective for inclusion as PS is of hydrophobic nature excluding the probability of solution swelling and the insertion of the metal ions. The PEO-cationic chemical coordination chemistry improves the incorporation process through chemical bonding of included ions [47]. The UV/ozone treatment removes any residual solvent and organic components, and cross-links and oxidizes the metal ions simultaneously. In the UV/Ozone treatment, ozone, an active oxidizing agent, is generated in situ from atmospheric oxygen by exposure to 185 nm UV light. The ozone produced subsequently photo dissociates into molecular oxygen and atomic oxygen upon exposure to 254 nm light. The latter specie reacts with the polymer to form free radicals and activated species that eventually remove organic portions of the polymer in the form of carbon dioxide, water, and a small amount of volatile organic compounds. Figure 6 shows the AFM, SEM and cross-sectional TEM images of well-ordered iron oxide nanodots formed after the UV/ozone treatment. The measured average centre-to-centre nanodot spacing remains unchanged as seen from the AFM and SEM images as these are formed via direct templating of the PS-*b*-PEO film. Figure 6a (AFM) and 6b (SEM) show iron oxide nanodots of uniform diameter of ~21 nm. The structural arrangement and interfaces with the substrate were analysed further by cross-sectional TEM (Figure 6c). The cross-sectional TEM image shows well-separated nanodots. Limited numbers of defects or cracks were observed. The hemispherical type structure of the nanodots is seen for all those imaged.

These iron oxide nanodots were then used as a hard mask for the formation of mesoporous silica nanopillars by pattern transfer into the MPS film. Briefly, a rapid silica etch process was used to remove the exposed mesoporous silica layer at the substrate surface whilst the layer below the iron oxide nanodots (mask) remained unaffected. This process results in the formation of mesoporous silica nanopillars with iron oxide at their uppermost surface. The top-down SEM image (Figure 7a) demonstrates a densely packed, uniform, ordered arrangement of pores over large areas after the pattern transfer. The high resolution SEM image also reveals that the hexagonally ordered pillars have an average diameter of ~21 nm at a spacing of ~42 nm. This implies that the etching does not damage the original pattern to any extent. The average height of the mesoporous silica nanopillars is found to be around 25 nm and mesoporous silica thin film thickness was estimated at 60 nm (measured from the cross-sectional TEM image shown in Figure 7b) after a silica etch for 10 s. These data clearly show that the mesoporous surface is robust enough to survive during etching.

## 4. Conclusions

A simple, generic and cost-effective route was demonstrated to synthesize 2D mesoporous silica thin films over wafer scale dimensions. A morphological study showed that the mesoporous silica thin film has hexagonally arranged pores with uniform pore diameter. The dimensional and structural ordering can be altered by varying the amount of silica precursor and different molecular weight of the surfactants. Lithographic resist and in situ hard mask block copolymer approaches were utilized followed by ICP dry etching to fabricate mesoporous silica channels. In comparison, BCP approach leads to lower dimension, high aspect ratio MPS channels. The width of the channels can be varied by using variety of commercially available lithographic resists whereas depth of the mesoporous silica channels varied by varying the etch time. Large area ordered mesoporous silica nanopillar arrays with smooth vertical sidewall profiles can be fabricated using an in situ hard mask block copolymer approach. The width and height of the nanopillars could be precisely varied depending on the diameter of the nanodots and the etching time respectively without altering their shape. The MPS channels or nanopillars arrays has a good coverage throughout the wafer scale area with direct access to the pore structures.

## Data Availability

Not applicable.

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
