# Peer review of "Fabrication of Dimensional and Structural Controlled Open Pore, Mesoporous Silica Topographies on a Substrate"

_nanomaterials, 2022, doi:10.3390/nano12132223_

Round 1

Reviewer 1 Report

In this work, Ghoshal et al., prepare mesoporous silica (MPS) thin films with different pore dimension, order and thickness depending on the silica precursor concentration and the molecular weight of the templating agent. Afterwards, channels are achieved on the MPS films by means of a lithographic mask or a block copolymer (BCP) mask obtained in situ.

In general, the study is interesting, but the authors should emphasize more the novelty of their work. While the lithographic technique appears to be quite “traditional”, the in situ BCP approach seems to be innovative and promising (although pretty complex to perform). I suggest the authors to clearly state what’s innovative in their approach compared to the literature, underlying the improvements/added values their approach can provide. Moreover, I wonder down to which size can a lithographic mask arrive? Would it be possible to obtain with a lithographic mask a pattern with a dimension comparable to that achieved by BCP approach?

In addition, here is a list of secondary comments:

1.       line 78: I suggest to replace “we have prepared” with an impersonal form such as “…were prepared”. Check throughout the manuscript.

2.       line 84: I imagine “rigorous stirring” was intended to be “vigorous stirring”.

3.       line 85: a verb is missing “silicon substrate WERE then calcined”.

4.       Paragraph 2.3: please add some references regarding the lithographic technique.

5.       Line 97: why strong agitation is applied when the film is developed? It may be useful to clarify it to the reader.

6.       Paragraph 2.4: I would suggest to add an explanatory scheme (similar to scheme 2). Such a scheme is presented in scheme £, but I don’t understand why this is placed in the results section instead of the experimental one.

7.       Table 1: it should be reported somewhere in the text before the table how the film thickness was measured and how the pore diameter was calculated (or measured?).

8.       Figure 2: when printed in b/w the curves cannot be distinguished clearly.

9.       Line 194-196: the explanation of the poor ordering is not clear. Please enrich and rephrase it.

10.   Figure 4: the scale bar unit should be “μm” and not “μΜ”.

11.   Figure 4: I suggest to add some arrows to indicate the different parts present in the picture (i.e., mask, etched silica…).

12.   Line 284-286: the sentence “following UV/ozone treatment…simultaneously” is not clear to me. Please, rephrase it.

13.   Line 331: how can the authors state that the samples exhibit large surface areas with direct access to pores? Did they measure it? If so, why isn’t the result reported?

Author Response

Reviewer 1:

In this work, Ghoshal et al., prepare mesoporous silica (MPS) thin films with different pore dimension, order and thickness depending on the silica precursor concentration and the molecular weight of the templating agent. Afterwards, channels are achieved on the MPS films by means of a lithographic mask or a block copolymer (BCP) mask obtained in situ.

In general, the study is interesting, but the authors should emphasize more the novelty of their work. While the lithographic technique appears to be quite “traditional”, the in situ BCP approach seems to be innovative and promising (although pretty complex to perform). I suggest the authors to clearly state what’s innovative in their approach compared to the literature, underlying the improvements/added values their approach can provide. Moreover, I wonder down to which size can a lithographic mask arrive? Would it be possible to obtain with a lithographic mask a pattern with a dimension comparable to that achieved by BCP approach?

Answer: The continual reduction of critical dimensions of advanced electronic devices challenges conventional ultraviolet (UV) lithography and requires the use of new and alternative patterning techniques such as double or triple patterning to create substrate features for use in both logic and memory device and interconnect level circuitry which proven to be slow, complex and expensive. While the lithographic mask can create features down to few tens of nanometers, BCP lithographic technique is capable to create feature as small as sub-5 nm depending on the type and molecular weights of BCPs.

The successful integration of BCP methods into the device requires ultimate control of the self-assembly and also the pattern transfer onto the underlying material. However, etching becomes complicated when feature size reduce and etch limitations of BCP patterns can lead to silicon features of low aspect ratio and high line edge roughness (LER). To overcome this barrier, we have generated a ‘hard mask’ material using our established insitu inclusion method.

We have included the explanations with references in the section 3.4.in the revised version of the manuscript.

In addition, here is a list of secondary comments:

Comment 1.  Line 78: I suggest to replace “we have prepared” with an impersonal form such as “…were prepared”. Check throughout the manuscript.

Answer: These are corrected in the revised version of the manuscript.

Comment 2. Line 84: I imagine “rigorous stirring” was intended to be “vigorous stirring”.

Answer: This is corrected in the revised version of the manuscript.

Comment 3. Line 85: a verb is missing “silicon substrate WERE then calcined”.

Answer: This is corrected in the revised version of the manuscript.

Comment 4.    Paragraph 2.3: please add some references regarding the lithographic technique.

Answer: We have included few references regarding the lithographic techniques in the ‘Introduction’ part of the revised manuscript.

Comment 5.  Line 97: why strong agitation is applied when the film is developed? It may be useful to clarify it to the reader.

Answer: Strong agitation was applied while developing thick film structures to improve the etching quality which reduces anisotropic etching and surface roughness and simultaneously increases the etching rate. The explanation is included in the section 2.3 in the revised manuscript.

Comment 6. Paragraph 2.4: I would suggest to add an explanatory scheme (similar to scheme 2). Such a scheme is presented in scheme £, but I don’t understand why this is placed in the results section instead of the experimental one.

Answer: We have added an explanatory scheme 3 in the revised version of the manuscript. Also, we have placed the description of Scheme 4 in the ‘Experimental section’ of the revised manuscript.

Comment 7.  Table 1: it should be reported somewhere in the text before the table how the film thickness was measured and how the pore diameter was calculated (or measured?).

Answer: The details of the thickness measurement instrument is included in section 2.6 and the method to calculate of the pore diameter is described in section 3.1 in the revised manuscript.

Comment 8. Figure 2: when printed in b/w the curves cannot be distinguished clearly.

Answer: We have applied different line styles for P123, P85 and P65 so that they can be easily distinguished when printed in b/w.

Comment 9.  Line 194-196: the explanation of the poor ordering is not clear. Please enrich and rephrase it.

Answer: Poor ordering and defects (shape, size and spacings between the pores) can be related to a number of reasons. These includes non-uniform width of the trench throughout the depth, incommensurability of the pore spacing and complex interfacial interactions of the trench wall with the precursor material during pore development. The explanation is included in section 3.2 in the revised version of the manuscript.

Comment 10.   Figure 4: the scale bar unit should be “μm” and not “μΜ”.

Answer: This is corrected in the revised version of the manuscript.

Comment 11.  Figure 4: I suggest to add some arrows to indicate the different parts present in the picture (i.e., mask, etched silica…).

Answer: We have included arrows and labels in Figure 4 in the revised version of the manuscript.

Comment 12.  Line 284-286: the sentence “following UV/ozone treatment…simultaneously” is not clear to me. Please, rephrase it.

Answer: we have rephrased and the explanation is included in section 3.4 in the revised version of the manuscript.

Comment 13.   Line 331: how can the authors state that the samples exhibit large surface areas with direct access to pores? Did they measure it? If so, why isn’t the result reported?

Answer: We have rephrased the sentence as ‘The MPS channels or nanopillars arrays has a good coverage throughout the wafer scale area with direct access to the pore structures’ and placed at the end of the ‘Conclusion’ section of the revised manuscript. The abstract is also modified accordingly.

Reviewer 2 Report

The study is devoted to preparing mesoporous silica thin films on a substrate surface. The obtained thin films were characterized utilizing the XRD, SEM, TEM, and AFM techniques. The paper is excellently written using good style and literacy. I recommend this well-prepared manuscript be accepted for publishing in Nanomaterials after the following minor revisions:

1. Unlike the Conclusions in the Abstract, the authors use Present Simple Passive instead of Present Perfect Passive or Past Simple Passive. Check it, please. Is it grammatically correct?

2. Line 75: Please, format the spelling of the units of molar mass.

3. Line 111: Please, check the spelling of the hydrate's formula.

4. Line 268: Please, move the dot before the reference.

5. Line 289: Please, correct the spelling in this line.

Author Response

The study is devoted to preparing mesoporous silica thin films on a substrate surface. The obtained thin films were characterized utilizing the XRD, SEM, TEM, and AFM techniques. The paper is excellently written using good style and literacy. I recommend this well-prepared manuscript be accepted for publishing in Nanomaterials after the following minor revisions:

Comment 1. Unlike the Conclusions in the Abstract, the authors use Present Simple Passive instead of Present Perfect Passive or Past Simple Passive. Check it, please. Is it grammatically correct?

Answer: This is corrected in the revised version of the manuscript.

Comment 2. Line 75: Please, format the spelling of the units of molar mass.

Answer: We have corrected this in the revised version of the manuscript.

Comment 3. Line 111: Please, check the spelling of the hydrate's formula.

Answer: We have corrected this in the revised version of the manuscript.

Comment 4. Line 268: Please, move the dot before the reference.

Answer: According to the journal format, the dot should be placed after the reference.

Comment 5. Line 289: Please, correct the spelling in this line.

Answer: This is corrected in the revised version of the manuscript.

Reviewer 3 Report

This paper reports a new method to synthesize 2D open pore structure silica material. It has scientific merit. The manuscript is well-organized and in excellent order.  

Author Response

This paper reports a new method to synthesize 2D open pore structure silica material. It has scientific merit. The manuscript is well-organized and in excellent order.  

Answer: We thank the reviewer for his valuable comments and time.